# Study on NGF and VEGF during the Equine Perinatal Period—Part 1: Healthy Foals Born from Normal Pregnancy and Parturition

**DOI:** 10.3390/vetsci9090451

**Published:** 2022-08-23

**Authors:** Nicola Ellero, Aliai Lanci, Vito Antonio Baldassarro, Giuseppe Alastra, Jole Mariella, Maura Cescatti, Luciana Giardino, Carolina Castagnetti

**Affiliations:** 1Department of Veterinary Medical Sciences (DIMEVET), University of Bologna, 40064 Bologna, Italy; 2Health Science and Technologies Interdepartmental Center for Industrial Research (HST-ICIR), University of Bologna, 40064 Bologna, Italy; 3IRET Foundation, 40064 Bologna, Italy

**Keywords:** neonatal foal, pregnancy, parturition, equine perinatal period, amniotic fluid, umbilical cord vein, placenta, nerve growth factor, vascular endothelial growth factor, brain-derived neurotrophic factor, thyroid hormones

## Abstract

**Simple Summary:**

The end of pregnancy, the birth and the sudden need for the fetus to adapt to the extra-uterine environment make the perinatal period eventful in all species. Trophic factors, such as nerve growth factor and vascular endothelial growth factor, as well as thyroid hormones, take part in the processes associated with the final maturation of the fetus. Our aim is to evaluate the levels of trophic factors and thyroid hormones obtained at parturition from a population of healthy mares and foals and in the first 72 h of foal life, as well as the expression of trophic factors in fetal membranes. The levels of both trophic factors decreased over time in foal plasma and a positive correlation was found between their levels at each time point, but no correlation was found with the thyroid hormone levels. Vascular endothelial growth factor was expressed in all fetal membranes, while nerve growth factor and its receptors were not expressed in the amnion. The close relationship between the two trophic factors in foal plasma over time and their fine expression in placental tissues appear to be key regulators of fetal development and adaptation to extra-uterine life.

**Abstract:**

The importance of trophic factors, such as nerve growth factor (NGF), vascular endothelial growth factor (VEGF), and brain-derived neurotrophic factor (BDNF) during the perinatal period, is now emerging. Through their functional activities of neurogenesis and angiogenesis, they play a key role in the final maturation of the nervous and vascular systems. The present study aims to: (i) evaluate the NGF and VEGF levels obtained at parturition from the mare, foal and umbilical cord vein plasma, as well as in amniotic fluid; (ii) evaluate NGF and VEGF content in the plasma of healthy foals during the first 72 h of life (T0, T24 and T72); (iii) evaluate NGF and VEGF levels at parturition in relation to the selected mares’ and foals’ clinical parameters; (iv) evaluate the relationship between the two trophic factors and the thyroid hormone levels (TT3 and TT4) in the first 72 h of life; (v) assess mRNA expression of NGF, VEGF and BDNF and their cell surface receptors in the placenta. Fourteen Standardbred healthy foals born from mares with normal pregnancies and parturitions were included in the study. The dosage of NGF and VEGF levels was performed using commercial ELISA kits, whereas NGF, VEGF and BDNF placental gene expression was performed using semi-quantitative real-time PCR. In foal plasma, both NGF and VEGF levels decreased significantly over time, from T0 to T24 (*p* = 0.0066 for NGF; *p* < 0.0001 for VEGF) and from T0 to T72 (*p* = 0.0179 for NGF; *p* = 0.0016 for VEGF). In foal serum, TT3 levels increased significantly over time from T0 to T24 (*p* = 0.0058) and from T0 to T72 (*p* = 0.0013), whereas TT4 levels decreased significantly over time from T0 to T24 (*p* = 0.0201) and from T0 to T72 (*p* < 0.0001). A positive correlation was found in the levels of NGF and VEGF in foal plasma at each time point (*p* = 0.0115; r = 0.2862). A positive correlation was found between NGF levels in the foal plasma at T0 and lactate (*p* = 0.0359; r = 0.5634) as well as between VEGF levels in the foal plasma at T0 and creatine kinase (*p* = 0.0459; r = 0.5407). VEGF was expressed in all fetal membranes, whereas NGF and its receptors were not expressed in the amnion. The close relationship between the two trophic factors in foal plasma over time and their fine expression in placental tissues appear to be key regulators of fetal development and adaptation to extra-uterine life.

## 1. Introduction

The end of pregnancy, the birth and the sudden need for the fetus to adapt to the extra-uterine environment make the perinatal period eventful in all species. The many functional and developmental changes that occur during this period are probably greater than in any other stage of life and the final maturation of the nervous and vascular systems plays a key role in the adaptation of the fetus to extra-uterine life.

Nerve growth factor (NGF) gained scientific preeminence as the founding and best-characterized member of the neurotrophin family. Neurotrophins, including NGF and brain-derived neurotrophic factor (BDNF), were originally indicated as neuroprotective factors, due to their effect in reducing apoptosis and promoting the survival and maintenance of specific populations of neurons in both the peripheral and central nervous systems during pre- and post-natal brain development. They are important for axon growth during development [1], neuronal function [2], developmental maturity of the cerebral cortex and synaptic plasticity, leading to the refinement of the connections [3], morphologic differentiation and neurotransmitter expression [4]. The biological functions of the neurotrophins are mediated through two classes of cell surface receptors, the tyrosine kinase receptors (TRK) and the p75 neurotrophin receptors (p75NTR). NGF sends its survival signals through activation of TRKA and can induce cell death by binding to p75NTR [5]. BDNF, via the corresponding TRKB receptor, is primarily present in immune cells, such as T cells and macrophages/microglia, and the number of BDNF-immunoreactive cells correlates well with lesion demyelinating activity [6]. Several other functional activities have, however, been attributed to neurotrophins and specifically to NGF, as suggested by NGF synthesis and/or expression of the high-affinity TRKA receptor in many cell types other than neurons, such as immune [7] and endocrine cells [8], endothelial cells and keratinocytes [9], and cardiomyocytes [10].

Although the equine NGF sequence has recently been identified in peripheral blood cells, the identification of NGF in the equine perinatal period has never been reported. In the study performed by Amagai et al. [11], there were no polymorphisms among the samples analyzed, and the neurotrophin showed more than 90% homology to human, mouse, rat, dog and bovine NGF, indicating that NGF-encoding is a highly conserved gene [11].

Vascular endothelial growth factor (VEGF) is a potent mitogen, morphogen and chemo-attractant for endothelial cells and is widely recognized as the most potent stimulator of vasculogenesis and angiogenesis [12]. VEGF, and its two main receptor molecules, fms-like tyrosine kinase 1 (FLT1; VEGFR1) and kinase insert-domain containing receptor (KDR; VEGFR2), have been shown to be expressed in the endometrium and placenta of mares during pregnancy [13]. Although originally described as a key angiogenic factor, it is now well established that VEGF also plays a crucial role in the nervous system. FLT1 has a weak tyrosine kinase activity, stimulates postnatal angiogenesis through intracellular signaling and evidence is now emerging that it also exerts neuroprotective effects [14]. KDR has strong tyrosine kinase activity, stimulates vascular permeability and stimulates survival of various neural cell types in the nervous system [15].

The hypothalamus–pituitary–thyroid (HPT) axis has specific functions, mostly related to metabolic activities, cell differentiation and development, but thyroid hormones also take part in processes associated with the central nervous system. From the beginning of fetal life, they regulate and stimulate the proliferation and growth of neurons, synaptic formation and myelination [16]. Although a possible interaction of thyroid hormones with members of the neurotrophin family and their functional receptors has been suggested [17], to the authors’ knowledge, the relationship between trophic factors and thyroid hormones in the equine perinatal period has never been investigated.

The biological nature of equine NGF and VEGF at foaling and in the early stages of equine neonatal life under physiological conditions has not yet been clarified. The aims of the present investigation, as a pilot study, are: (i) to evaluate NGF and VEGF levels in plasma samples obtained at parturition from the mare’s jugular vein, the foal’s jugular vein and the umbilical cord vein, as well as in amniotic fluid; (ii) to evaluate NGF and VEGF content in the plasma of healthy foals during the first 72 h of life; (iii) to evaluate the NGF and VEGF levels at parturition/birth in relation to the selected mares’ and foals’ clinical parameters; (iv) to evaluate the relationship between the two trophic factors and the thyroid hormone levels (TT3 and TT4) in the first 72 h of life; (v) to assess the mRNA expression of NGF, VEGF and BDNF and their cell surface receptors (p75 neurotrophin receptor and tropomyosin receptor kinase A for NGF; kinase insert domain receptor and fms-related receptor tyrosine kinase 1 for VEGF; tropomyosin receptor kinase B for BDNF) in the placenta. This study is based on the hypothesis that NGF and VEGF, as well as signaling through their specific receptors and interaction with the HPT axis, represent key neuroprotective and angiogenic factors in fetal adaptation to extra-uterine life. The first part of the present study focuses on understanding the biological significance of their expression under physiological conditions for the regulation of pre- and post-natal development in the equine species. The levels of trophic factors and thyroid hormones at the time of birth and in the first 72 h of life well represent the transition of the healthy equine neonate to extra-uterine life.

## 2. Materials and Methods

### 2.1. Population

Fourteen Standardbred healthy foals born from healthy mares with normal pregnancy and parturition hospitalized at the Perinatology and Reproduction Unit (Equine Clinical Service, Department of Veterinary Medical Sciences) of the University of Bologna during the 2018 to 2021 foaling seasons were included in the study.

The mares were hospitalized at about 310 days of pregnancy because the owners requested an attended parturition. They were housed in separate wide straw-bedded boxes, fed hay ad libitum and concentrates twice a day, and were allowed to go to pasture during the day. At admission, a complete clinical evaluation, including complete blood count (ADVIA 2120 analyzer, Siemens Healthcare srl, Milan, Italy) and transrectal ultrasonography, were performed. Subsequently, the mares were clinically evaluated twice a day and by ultrasonography every 10 days until parturition. After delivery, macroscopic and histopathological examination of the placenta was performed in all mares.

Mares with a diagnosis of high-risk pregnancy [18,19], dystocia [20] or placental insufficiency [21,22] were excluded from the study.

At birth, a complete clinical evaluation, including complete blood count and serum biochemistry (AU 400 analyzer, Olympus/Beckman Coulter, Lismeehan, Ireland), was performed in all foals. The foals were monitored throughout the hospitalization period by a clinical examination performed every 6 h.

The foals born from normal pregnancy and birth were classified as healthy when they had an Apgar score ≥ 9 [23] and a normal clinical evaluation during the course of hospitalization, including a complete blood count and serum biochemistry at birth and an IgG serum concentration > 800 mg/dL (by immunoturbidimetric method; DVM Rapid Test II, MAI Animal Health, Elmwood, WI, USA) at 24 h of life.

The foals classified as affected by neonatal disease at birth or during the hospitalization period were excluded from the study.

### 2.2. Clinical Data and Sample Collection

The following data were recorded for each mare: age (years), parity, gestation length (days), length of stage II parturition (min), placenta/foal weight ratio (%), macroscopic and histopathological evaluation of the placenta.

The following data were recorded for each foal: sex, weight (kg), Apgar score at birth [23], blood glucose at birth (mg/dL) (Medisense Optium, Abbott Laboratories Medisense Products, Bedford, MA, USA), umbilical cord vein (UV) and jugular vein (JV) lactate concentrations at birth (mmol/L) (Lactate SCOUT+, Leipzig, Germany) as well as hematobiochemical parameters.

All biological samples were harvested as part of the clinical program of peripartum monitoring; owners gave written consent to use samples for research.

The amniotic fluid (AF) was collected within 5 min of the appearance of the amniotic sac, through the vulva by needle puncture of the amnion, using a 60 mL sterile syringe. The samples were immediately stored at −20 °C and analyzed within 6 months.

As soon as the foal was born, the umbilical structures were identified both visually and via manual palpation. The UV sample was taken as close to the foal’s body wall as possible, using a 21 g butterfly needle attached to a vacuum system as well as K3 EDTA and serum (clot activator) tubes (Vacutest Kima, Arzergrande, Italy).

Blood was collected in K3 EDTA and serum tubes by jugular venipuncture in all mares at parturition (TP) and in all foals at each time point (at birth: T0; at 24 h of life: T24; at 72 h of life: T72). All blood samples were centrifuged at 3000 g for 10 min at 4°C and aliquots of supernatant plasma/serum were collected, immediately stored at −20 °C and analyzed within 6 months.

Immediately after expulsion, the fetal membranes were weighed and subsequently placed on their side in an F shape to perform a macroscopic evaluation. To ensure an appropriate comparison between the different subjects, one placenta sample that was uniform in size (2 × 2 cm) was collected from each area of fetal membranes segments: body, pregnant horn, nonpregnant horn, cervical pole and amnion [24]. Samples were formalin-fixed and paraffin-embedded. Routine histological hematoxylin- and eosin-stained slides were obtained.

For molecular biology investigations, in 8 of the 14 mares, two 0.5 × 0.5 cm samples were collected from the amnion and the body of the placenta at the base of the pregnant horn, near the umbilical cord attachment. The latter was manually separated into two portions: chorion and allantois. Samples were washed with sterile saline, frozen in liquid nitrogen for 60 s, stored at −80 °C and analyzed within 6 months.

### 2.3. Measurement of NGF and VEGF by ELISA

Dosage of NGF and VEGF levels was performed using commercial ELISA kits (MyBiosource, San Diego, CA, USA), which detect equine NGF and VEGF (Horse NGF/VEGF ELISA; NGF, Cod. MBS040618; VEGF, Cod. MBS035093). All samples were centrifuged at 4000 g for 10 min and at 4 °C prior to the assay and then analyzed according to the manufacturer’s indications. Briefly, undiluted samples and standard curves were added to the wells and immediately mixed with HRP-conjugate. After 60 min of incubation at 37 °C, the wells were washed four times and two chromogens were added in sequence. The stop solution was added after a second incubation (15 min at 37 °C) and the plate read within 15 min at 450 nm. Optical density values were interpolated on a linear standard curve using GraphPad Prism v 6.0. NGF standard curve ranges from 15.6 ng/mL to 500 ng/mL, whereas VEGF standard curve ranges from 31.2 pg/mL to 1000 pg/mL.

### 2.4. Measurement of Thyroid Hormones

Basal thyroid hormone levels of total triiodothyronine (TT3) and total thyroxine (TT4) from the mare JV, UV and the foal JV at T0, T24 and T72 were determined using the Siemen’s Immulite TT3 and TT4 kits (Immulite Canine TT3 and TT4; Siemens, Oakville, ON, Canada) validated for use with equine serum at the Endocrine Laboratory, Prairie Diagnostic Services, Saskatoon, SK [25,26]. The analytical sensitivity of the TT3 Immulite assay is 0.54 nmol/L. Specificity data from the manufacturer regarding the anti-TT3 antibody identified a 100% cross-reaction with triiodo-L-thyronine, 100% with triiodo-D-thyronine, 1.3% with tetraiodothyroacetic acid and 0.7% with triiodothyroacetic acid. The analytical sensitivity of the TT4 assay was 0.15 nmol/L. Specificity data regarding the anti-TT4 antibody from the manufacturer identified a 100% cross-reactivity with L-thyroxine, 55% with D-thyroxine, 16% with tetraiodothyroacetic acid and 3.2% with triiodo-L-thyronine.

### 2.5. Placental Gene Expression

Chorion, allantois and amnion tissues were homogenized, and total RNA isolation was performed using RNeasy Microarray Tissue Mini Kit (Qiagen, Hilden, Germany, Cod. 73404) by the automated extractor QIAcube Connect (Qiagen).

Total RNA was eluted in RNase Free Water, and using a spectrophotometer (Nanodrop 2000, Thermo Scientific, Waltham, MA, USA, absorbance values at 260, 280 and 320 nm were measured. For the reverse transcription to generate the cDNA, the iScript™ gDNA Clear cDNA kit (Biorad, Hercules, CA, USA, Cod. 1725035BUN) was used, while semi-quantitative real-time PCR was performed using the CFX96 real-time PCR system (BioRad, Hercules, CA, USA). The reactions were performed in a final volume of 20 µL consisting of SYBR Green qPCR master mix (BioRad, Cod. 1725274), 0.4 µM forward and reverse primers and nuclease-free water. The no-RT control was processed in parallel with the others and tested by real-time PCR for every primer pair. No template controls were added for each gene expression analysis. All primers were designed using Primer Blast software (NCBI, Bethesda, MD, USA) and synthesized by IDT (Coralville, IA, USA). Specific sequences of primers are listed in Table 1.

Thermal profile of PCR reactions consisted first of a denaturation step (98 °C, 3 min) and 40 cycles of amplification (95 °C for 10 s and 60 °C for 1 min). At the end of the amplification cycles, the dissociation curve was obtained by following a procedure consisting of first incubating samples at 95 °C for 1 min to denature the PCR-amplified products, then ramping temperature down to 65 °C and finally increasing temperature from 65 °C to 95 °C at the rate of 0.5 °C/s, continuously collecting fluorescence intensity over the temperature ramp.

### 2.6. Statistical Analysis

The one-way ANOVA test was used to evaluate significant differences in biomarker levels at different collection times. Matched multiple comparison versus T0 was performed.

To evaluate correlations between parameters, Pearson or Spearman correlation coefficients were calculated, with Gaussian or non-Gaussian distribution, respectively.

NGF and VEGF levels were correlated in the foal’s JV plasma, between mares and foals and with the data recorded for each mare at TP (age, parity, gestation length and placenta/foal weight ratio) and the following clinical and hematobiochemical parameters were recorded for each foal at T0: weight at birth, UV and JV lactate concentration, serum creatine kinase, total bilirubin, blood urea nitrogen, creatinine, magnesium and serum amyloid A.

A *p* < 0.05 was considered statistically significant. All statistical analyses were carried out using commercial software GraphPad Prism version 6.00.

## 3. Results

### 3.1. Population Characterization

The clinical data collected from the mares and foals included in this study are shown in Table 2. The clinical examinations performed during the hospitalization period were within normal limits at all time points, with all foals exhibiting normal immediate post-foaling behavior, including the ability to stand, nurse and pass meconium and urine. 

The results of the complete blood count and serum biochemistry, including JV glucose and UV and JV lactate at T0 (measured through rapid methods) are summarized in Appendix A [27,28,29,30] in the Appendix A. Macroscopic and histopathological evaluations of the placenta were also within normal limits in all mares.

### 3.2. Biomarkers (NGF, VEGF, TT3 and TT4) in Biological Fluids

NGF, VEGF, TT3 and TT4 were dosed in JV at each time point analyzed (0, 24 and 72 h from birth). Trends in plasma levels of NGF and VEGF and serum levels of TT3 and TT4 in foals in the first 72 h of life are illustrated in Figure 1. In foal plasma, both NGF and VEGF levels decreased significantly over time, from T0 to T24 (*p* = 0.0066 for NGF; *p* < 0.0001 for VEGF) and from T0 to T72 (*p* = 0.0179 for NGF; *p* = 0.0016 for VEGF). In foal serum, the TT3 levels increased significantly over time, from T0 to T24 (*p* = 0.0058) and from T0 to T72 (*p* = 0.0013), whereas TT4 levels decreased significantly over time, from T0 to T24 (*p* = 0.0201) and from T0 to T72 (*p* < 0.0001).

Furthermore, as shown in Figure 2, a positive correlation was found in the plasma levels of NGF and VEGF in the foal’s JV at each time point (*p* = 0.0115; r = 0.2862).

Both NGF and VEGF were also dosed in all other matrices analyzed: amniotic fluid, plasma obtained from umbilical cord vein and plasma obtained from the mare’s jugular vein at parturition. Overall, the results are summarized in Table 3.

No correlations were found between the NGF and VEGF levels in the amniotic fluid, the plasma obtained from the umbilical cord vein and the plasma obtained from the mare’s jugular vein at parturition. Moreover, no correlation between NGF and VEGF mares and foals levels was found.

### 3.3. Equine NGF/VEGF and Clinical Data

As show in Figure 3, a positive correlation was found between NGF levels in the foal’s JV at T0 and the lactate concentration at T0 (*p* = 0.0359; r = 0.5634) and between VEGF levels in the foal’s JV at T0 and the serum creatine kinase at T0 (*p* = 0.0459; r = 0.5407).

### 3.4. Equine NGF, VEGF, BDNF and Their Receptors Gene Expression in the Placenta

The gene expression of NGF, VEGF and BDNF is shown in Table 4. In the chorion, the three neurotrophins NGF, VEGF and BDNF were expressed, as well as NGF and VEGF receptors, while only the BDNF receptor TRKB was not expressed. Additionally, in the allantois, the neurotrophins were all expressed, together with the NGF low-affinity receptor p75NTR and the two VEGF receptors KDR and FLT1, while the high-affinity NGF receptor TRKA and the BDNF receptor TRKB resulted not detectable. In the amnion, VEGF and both its receptors KDR and FLT1 were expressed, as well as BDNF; however, NGF and both its receptors p75NTR and TRKA, together with the BDNF receptor TRKB, were not expressed.

## 4. Discussion

Compared to the human species, little is known about the physiology of trophic factors in the equine perinatal period. NGF and VEGF are pleiotropic molecules which exert a wide variety of effects on different body districts and cell types, both during development and in adulthood. In the first part of this study, the NGF and VEGF levels were measured in the plasma of mares with normal pregnancy and parturition, in the amniotic fluid, in the umbilical vein and in the plasma of their healthy foals in the first 72 h of life. To obtain a more complete picture, the trend of serum thyroid hormones (TT3 and TT4) in the first 72 h of life and the gene expression of NGF, VEGF, BDNF and their receptors (TRKA, p75NTR, FLT-1, KDR, TRKB) in the fetal membranes were evaluated. The compartments explored in this study should reflect feto-placental physiology (fetal membranes, amniotic fluid and umbilical vein plasma) and the physiology of the mother/neonate (mare and foal plasma).

Higher amounts of NGF than VEGF were found in mare and foal plasma, umbilical vein plasma and amniotic fluid, but both trophic factors decreased significantly in the first 72 h of life in the plasma of the neonatal foals. This decline is most consistent for VEGF. Experimental and human clinical studies indicate that many body districts and cell types synthetize neurotrophic factors [7,8,9,10]; thus, it is reasonable to assume that circulating NGF and VEGF levels could be in part of fetal and neonatal origin. The most interesting aspect was perhaps the correlation that exists between the NGF and VEGF levels in the foal plasma at each time point. The close relationship between the two trophic factors has never been found in experimental or clinical studies before. The significance of this positive correlation should probably be investigated in light of the NGF and VEGF roles in the brain compartment. From a translational point of view, studies conducted on human perinatology showed that the blood levels of neurotrophins, including NGF and BDNF, are similar to those in the brain [31] and that the neural tube, from which the brain and spinal cord develop, becomes vascularized by a process involving VEGF [32]. In addition to its role in directing vessel sprouting, VEGF also regulates neuronal cell migration in the central nervous system in a mouse model [33].

Although the source of trophic factors in equine amniotic fluid is still unknow, explanatory fetal and placental mechanisms should be considered. Fetal urine can reasonably contain all molecules which cross the blood–brain barrier, including NGF, and changes to the brain can be reflected in urine [34]. Since fetal urine is a major component of the amniotic fluid during late gestation in equine species [35], it is reasonable to assume it may also be a source of NGF and VEGF in the amniotic fluid. This may seem especially true for NGF due to its lack of expression in the amnion.

The umbilical cord vein may contain blood components of placental and maternal origin; in fact, it is reasonable to assume that trophic factors such as NGF and VEGF cross the utero–placental barrier to reach the fetal compartment [36]. The lack of correlation between maternal, umbilical cord vein and neonatal plasma levels at parturition does not make it possible to confirm the possible transfer of these factors between the materno-placental and neonatal circulation. This result is probably related to the participation of the fetus in the regulation of these pathways.

The few studies which have been performed on the human perinatal period under physiological conditions are related to the amniotic fluid and the maternal/neonatal plasma levels. It has been reported that NGF in the human amniotic fluid increases with gestational age [37], and that NGF is the only neurotrophin correlated to birthweight [31]. The maternal and umbilical cord plasma NGF levels, as well as the VEGF levels in the amniotic fluid, were correlated with fetal growth, with likely implications for postnatal neurodevelopment [38,39]. In the present study, no correlations were observed between plasma NGF and VEGF levels and the selected mares’ and foals’ clinical parameters, probably due to the low number of samples and high individual variability. In contrast to the reports in human literature, NGF is not correlated to birthweight in the equine species [31]. The correlation of plasma levels of NGF and VEGF in the foal with hematobiochemical parameters at birth, however, showed significant indices. From a clinical perspective, NGF correlated positively with blood lactate concentrations in healthy neonatal foals. Lactate could be evaluated to monitor the postpartum period and to indicate the need for prompt intervention, since it is the end-product of both aerobic and anaerobic glucose metabolism [28]. Although the blood lactate concentration may be physiologically elevated in the first 12 h of life, hyperlactatemia is produced in the event of hypoxia and poor tissue perfusion [40]. VEGF correlated positively with the serum creatine kinase (CK) level; this enzyme is present in the heart, skeletal muscles and brain. A mild elevation of CK level at birth is natural in neonatal foals, and the possible causes of this elevation are pressure in the birth canal and mild muscle damage from reactive oxygen [41]. However, hypothetical and speculative conclusions concerning correlations with blood parameters need to be critically evaluated in future investigations, since none of these parameters were highly correlated.

Although the thyroid function of the fetus improves steadily from mid-gestation, maternal thyroid hormones are required until the end of the pregnancy. Under physiological conditions, this study showed the serum TT3 and TT4 levels were several times higher in neonate foals at birth than in their mares, but similar to those of the umbilical cord vein. They cross the placental barrier to enter the fetal blood, and subsequently the blood–brain barrier (BBB) and blood–cerebrospinal fluid barrier [16]. Transplacental transport is mediated by transporters in the cell membrane and thyroid hormone-binding proteins in trophoblast cells, and during passage through the BBB they are captured by endothelial cells. T4 then enters astrocytes and is converted into T3 and transported to neurons [16]. The increasing trend in TT3 levels recorded in foal serum in the first 72 h of life is in agreement with a previous study that used the same time sampling to investigate thyroid function [42]. In accordance with Pirrone et al. [42], the serum TT4 levels in foals decreased significantly in the first 72 h of life. Such high levels of thyroid hormones at birth can only be detected in neonate foals. They are higher than in any other species at any physiological stage and are responsible for their high thermogenic capacity and remarkable rapidity of growth during the perinatal period, especially of the nervous system [42]. From a translational perspective, several studies suggest the possible interaction of thyroid hormones with members of the neurotrophin family and their functional receptors, not only during brain maturation, but also during brain maintenance [17]. Although experimental studies have long since confirmed that thyroid hormones mediate direct effects on NGF-induced expression in the brain of neonatal mice [43] and in the cerebellum of perinatal rats [44], in the present study a relationship between trophic factors and thyroid hormones was not found in healthy foals.

Concerning fetal membranes, the mare exhibits a diffuse epitheliochorial placenta. At term, the extensive allantochorion consists of no more than a single thin layer of low columnar-to-cuboidal trophoblast cells which overlies and provides the essential structural framework for an incredibly densely packed mass of fetal capillaries supported in minimal amounts of allantoic mesoderm [13]. The amnion is the inner fetal membrane and consists of a cuboid epithelial layer that comes into direct contact with the amniotic fluid [45].

In the human species, NGF is reported to be present in the placenta [46] and mRNA expression has been demonstrated in the trophoblast, amnion/chorion and maternal decidua both early in gestation and at term [47]. In the equine species, NGF and its two main receptors, p75NTR and TRKA, are fully expressed in the chorion, whereas only the p75NTR receptor is expressed in the allantois. Surprisingly, they were not found to be expressed in the amnion, despite the NGF synthesis and/or expression of the high-affinity TRKA receptor in many cell types other than neurons, including endothelial cells [9]. In addition, a recent study also suggests that for a healthy human pregnancy, optimal NGF expression in the feto–maternal interface is essential [48], because it influences the process of angiogenesis as it exerts a potent angiogenic effect [49]. Prior to this, studies on neurotrophins in the placenta of domestic animals have never been conducted, and due to the profound differences between human and equine placenta, the authors cannot advance hypotheses on placental synthesis or excretion of NGF, but the main source of NGF in the amniotic fluid appears to be of fetal origin.

BDNF and its receptor TRKB were found in the uterus of many species, including the horse. They are co-expressed and co-localized in the glandular epithelium, luminal epithelium, vascular smooth muscle and myometrium of the mare during early pregnancy [50]. BDNF and TRKB have also been previously shown to activate the adhesion [51], angiogenesis [52], apoptosis [53] and proliferation [54] pathways, mainly in the brain and nervous system. Each of these pathways is also of paramount importance at the end of the pregnancy; however, little is known about the role of BDNF and TRKB in reproductive physiology. The results of the present study add limited but novel information on the topic. In the placenta of mares at term, BDNF was expressed in the chorion, allantois and amnios, whereas the TRKB receptor was not expressed in any of these portions. The lack of interaction between BDNF and the TRKB receptor observed in full-term mares may serve to inhibit the classic BDNF–TRKB pathways, and also prevent nerve growth into placental tissue, which is, at the time of birth, soon degraded and removed in a cyclical manner. While the literature that supports BDNF expression during human pregnancy, particularly in the brain, is growing [55], its specific function is still unclear, but these results suggest that this signaling pathway is potentially important in normal equine pregnancy physiology.

VEGF, and its two major receptor molecules FLT1 and KDR, seem to be the principal vasculogenic and angiogenic factors. They are localized throughout most of the gestation on the two principal secretory cell types of the equine placenta, the glandular and luminal epithelia of the maternal endometrium and the trophoblast of the fetal allantochorion [13,56]. Our results also indicate that in the full-term mare, VEGF and its receptors are well expressed in the amnion. FLT1 stimulates postnatal angiogenesis through intracellular signaling and also exerts neuroprotective effects [14]. KDR stimulates vascular permeability as well as the survival of various types of neural cells in the nervous system [15]. Studies on pregnant sheep have shown that amniotic VEGF expression is regulated in vivo simultaneously with increased intramembranous uptake, qualifying VEGF as a candidate factor to influence amnion permeability [57]. It may therefore be reasonably postulated that, in the mare and in the fetus, VEGF, FLT1 and KDR together facilitate the development of maternal and fetal vascular and nervous networks for the interchange of nutrients and stimuli.

Some limitations of the study design should be noted. Neurotrophin signaling and regulation is really complex: each receptor can bind more than one ligand with varying affinity, multiple splice and transcript variants of ligands and receptors exist, several post-translational modifications may be present, ligands are first translated as pro-proteins which bind receptors and ligands can exist as monomers or dimers. In addition, the circulating BDNF levels were not assessed in the population of mare and foal pairs but only in the placental tissues. Due to the lack of information on the physiology of trophic factors in the horses and the profound difference in terms of fetal development and placental structure between the equine and human species, references to other species are necessary to get a broader view of the topic.

## 5. Conclusions

The first part of the present study provided limited but novel insights into the role of NGF and VEGF in the equine perinatal period, considering that limited studies are available on humans as well. The close relationship between the two trophic factors in foal plasma over time and their fine expression in placental tissues are certainly key regulatory factors in fetal development and adaptation to extra-uterine life.

The novel information obtained from the population of healthy mares and foals will now allow us to analyze the two trophic factors in sick neonates.

## Figures and Tables

**Figure 1 vetsci-09-00451-f001:**
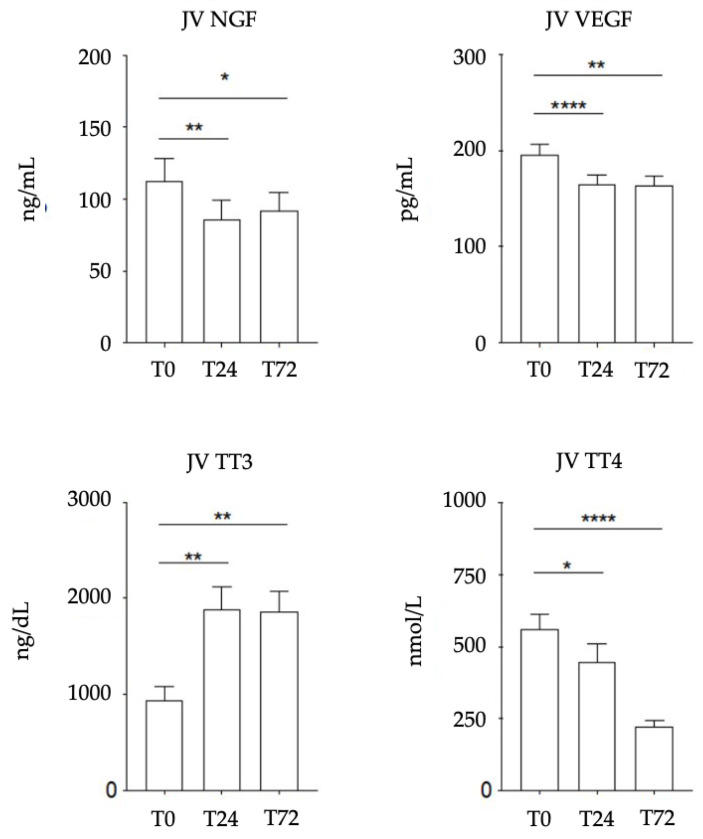
Time-dependent changes in plasma NGF–VEGF levels and serum TT3–TT4 levels in jugular vein (JV) of healthy foals. T0: birth; T24: 24 hours from birth; T72: 72 hours from birth. NGF, VEGF and TT4 levels decrease significantly from T0 to T24 and from T0 to T72, whereas TT3 levels increase significantly from T0 to T24 and from T0 to T72. Statistical analysis: one-way ANOVA, with the Geisser–Greenhouse correction. Adjusted *p* values JV NGF: * = 0.0179; ** = 0.0066; adjusted *p* values JV VEGF: ** = 0.0016; **** < 0.0001; adjusted *p* values JV TT3: ** = 0.0058; ** = 0.0013; adjusted *p* values JV TT4: * = 0.0201; **** < 0.0001.

**Figure 2 vetsci-09-00451-f002:**
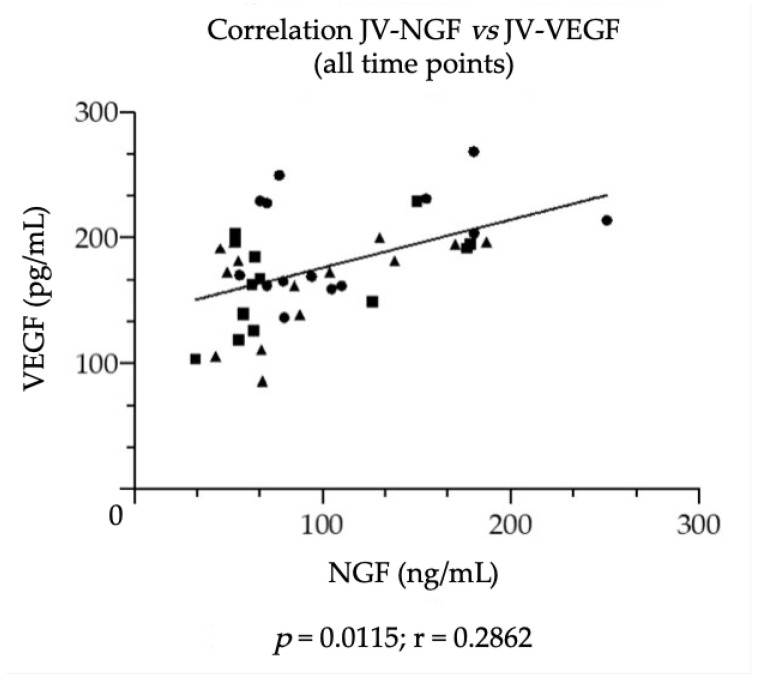
Correlation found between NGF and VEGF levels in plasma samples obtained from foal’s jugular vein at three consecutive time points (round points T0: birth; square points T24: 24 h from birth; triangular points T72: 72 h from birth). Statistical analysis: non-parametric Spearman correlation. *p* = 0.0015; r = 0.2862.

**Figure 3 vetsci-09-00451-f003:**
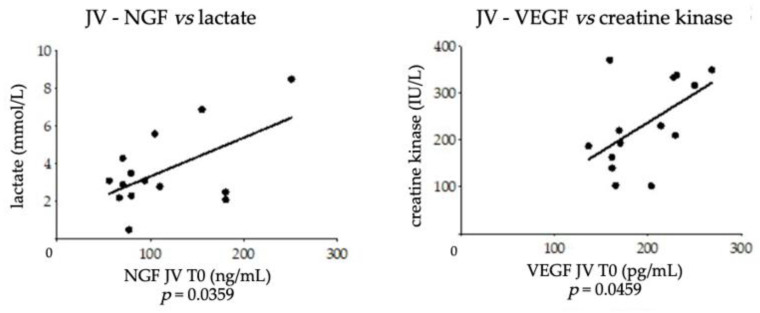
Significant correlations found between NGF and VEGF levels in plasma samples obtained from foal’s jugular vein and hematobiochemical parameters at T0. Statistical analysis: parametric Pearson correlation (JV–NGF vs. lactate and JV–VEGF vs. creatine kinase).

**Table 1 vetsci-09-00451-t001:** List of the gene-specific primer sequences.

Genes	Primer Sequences (5′–3′)
*NGF* *Nerve growth factor*	Forward: GGGCCCATTAACGGCTTTTC Reverse: CATTGCTCTCTGTGTGGGGT
*P75NTR* *p75 Neurotrophin receptor*	Forward: GAGCCAACCAGACTGTGTGT Reverse: GGTAGTAGCCATAGGCGCAG
*TRKA* *Tropomyosin receptor kinase A*	Forward: GGAGCTGAGAAACCTCACCAT Reverse: GCACAGGAACAGTCCAGAGG
*VEGF* *Vascular endothelial growth factor*	Forward: AACGACGAGGGCCTAGAGT Reverse: CAAGGCCCACAGGGATTTTCT
*KDR* *Kinase insert domain receptor*	Forward: GATGACAACCAGACGGACAGT Reverse: TTTTGCTGGGCATCAGTCCA
*FLT1* *Fms-related receptor tyrosine kinase 1*	Forward: CTGGCATCCCTGTAACCACA Reverse: AGGGTGCTAGCCGTCTTATTC
*BDNF* *Brain-derived neurotrophic factor*	Forward: CATGTCTATGAGGGTCCGGC Reverse: CATGTCCACTGCCGTCTTCT
*TRKB* *Tropomyosin receptor kinase B*	Forward: CGGGAACACCTCTCGGTCTA Reverse: CTGGACCAACACCTTGTCTTGA

**Table 2 vetsci-09-00451-t002:** Clinical data collected from mares and foals born from attended parturition. *n* = number of animals. Data are expressed as mean ± standard deviation (min–max).

Mare				Foal			
**Age**	Parity	Gestational Length	Stage II Labor Length	Sex	Weight	Apgar Score	Placenta/Foal Weight Ratio
(Years)		(Days)	(min)		(kg)	(0–10)	(%)
10.4 ± 5.1	3.8 ± 3.4	339.6 ± 9.2	11.9 ± 5.4	Males *n* = 5	48.2 ± 5.7	9.4 ± 0.6	10.5 ± 1.4
(5–20)	(1–12)	(325–355)	(6–25)	Females *n* = 9	(38–55)	(8–10)	(7.8–12.2)

**Table 3 vetsci-09-00451-t003:** NGF–VEGF levels in amniotic fluid and in plasma samples obtained from umbilical cord vein, mare’s jugular vein (JV) at parturition (TP) and foal’s jugular vein (JV) at three consecutive time points (T0: birth; T24: 24 h from birth; T72: 72 h from birth); TT3–TT4 levels in serum samples obtained from umbilical cord vein, mare’s jugular vein at parturition (TP) and foal’s jugular vein at three consecutive time points (T0: birth; T24: 24 h from birth; T72: 72 h from birth). *n* = number of dosed samples; NA = data not available. Data are expressed as mean ± standard deviation (min–max).

	Amniotic Fluid	Umbilical Cord Vein	Mare’s JV	Foal’s JV		
			TP	T0	T24	T72
NGF (ng/mL)	146.8 ± 43.6	134.2 ± 46.8	101.9 ± 67.8	112.7 ± 57.5	85.9 ± 49.6	91.9 ± 47.6
	(95.4–233.0)	(63.4–214.5)	(38.3–208.4)	(56.1–251.2)	(32.6–178.8)	(43.2–187.3)
	(*n* = 14)	(*n* = 13)	(*n* = 8)	(*n* = 14)	(*n* = 14)	(*n* = 14)
VEGF (pg/mL)	268.1 ± 17.63	192.6 ± 32.0	134.0 ± 26.3	196.3 ± 40.7	164.9 ± 36.9	163.9 ± 38.2
	(239.6–294.2)	(131.3–231.1)	(80.8–162.9)	(136.4–268.6)	(103.5–229.4)	(85.8–200.3)
	(*n* = 12)	(*n* = 13)	(*n* = 8)	(*n* = 14)	(*n* = 14)	(*n* = 14)
TT3 (ng/dL)	NA	409.2 ± 102.8	59.8 ± 15.5	936.2 ± 550.1	1888.1 ± 861.3	1858.6 ± 755.7
		(252.0–584.0)	(40.0–87.8)	(362.0–1828.0)	(552.0–3528.0)	(412.0–3128.0)
		(*n* = 12)	(*n* = 8)	(*n* = 14)	(*n* = 14)	(*n* = 13)
TT4 (nmol/L)	NA	585.9 ± 280.2	15.5 ± 4.4	563.6 ± 180.2	446.8 ± 230.8	223.1 ± 71.5
		(336.0–1180.0)	(12.9–20.6)	(292.0–848.0)	(198.0–951.0)	(97.7–363.0)
		(*n* = 12)	(*n* = 8)	(*n* = 14)	(*n* = 14)	(*n* = 13)

**Table 4 vetsci-09-00451-t004:** Gene expression of NGF, VEGF, BDNF and their receptors in the three portions of placenta: chorion, allantois and amnion. Data are shown as gene expressed (exp) or not expressed (ne). Genes were indicated as expressed (exp) if they were amplified with a Cq ≤ 37. *p75NTR* = p75 neurotrophin receptor; TRKA = tyrosine kinase receptor A; KDR = kinase insert domain containing receptor; FLT1 = fms-like tyrosine kinase 1; TRKB = tyrosine kinase receptor B.

	Ligand	Receptors		Ligand	Receptors		Ligand	Receptor
	*NGF*	*p75NTR*	*TRKA*	*VEGF*	*KDR*	*FLT1*	*BDNF*	*TRKB*
Chorion	exp	exp	exp	exp	exp	exp	exp	ne
Allantois	exp	exp	ne	exp	exp	exp	exp	ne
Amnion	ne	ne	ne	exp	exp	exp	exp	ne

## Data Availability

Not applicable.

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
