# Peer review of "Study on NGF and VEGF during the Equine Perinatal Period—Part 1: Healthy Foals Born from Normal Pregnancy and Parturition"

_vetsci, 2022, doi:10.3390/vetsci9090451_

Round 1

Reviewer 1 Report

The first part of data presented in the manuscript by Ellero and colleagues provides physiological parameters about the level of NGF and VEGF in the foal and mares concomitantly to the parturition time as well as in the first hours of foal life. The experimental design is robust and scientifically sound, and the results are well discussed. Some minor remarks:

- would be better to explicit why NGF and VEGF, as well as tyroid hormones, have been measured at birth, T24 and T72 in foals

- discussion on BDNF and TRKB expression in fetal membranes (lines 435-450) should be moved before discussion of VEGF (line 420 oward). Furthermore, Authors should better clarify the concept "and also prevent nerve growth into placental tissue" expressed in line 446.

Author Response

Responses to reviewers

Review 1

The first part of data presented in the manuscript by Ellero and colleagues provides physiological parameters about the level of NGF and VEGF in the foal and mares concomitantly to the parturition time as well as in the first hours of foal life. The experimental design is robust and scientifically sound, and the results are well discussed. Some minor remarks:

The authors thank the reviewer for his careful work and suggestions.

- would be better to explicit why NGF and VEGF, as well as tyroid hormones, have been measured at birth, T24 and T72 in foals.

Thanks for the suggestion. In the clinical study design, the levels of trophic factors and thyroid hormones at the time of birth (T0) and in the first 72 h of life (T24 and T72) well represent the transition of the healthy equine neonate to extra-uterine life. A sentence was added to the end of the Intro section, highlighted in the text.

- discussion on BDNF and TRKB expression in fetal membranes (lines 435-450) should be moved before discussion of VEGF (line 420 oward). Furthermore, Authors should better clarify the concept "and also prevent nerve growth into placental tissue" expressed in line 446.

Thanks for the suggestion. The part of discussion on BDNF in fetal membranes has been moved before the one on VEGF and the changes to the numbering of the references have been highlighted. In addition, the authors have tried to better express the required concept, expressed in line 446. The lack of interaction between BDNF and the TrkB receptor observed in full-term mares was unexpected. Although it is risky to provide an explanation for this result, it may serve to inhibit the classic BDNF-TRKB pathways, and also prevent nerve growth into placental tissue, which is, at the time of birth, soon degraded and removed in a cyclical manner.

Reviewer 2 Report

The authors presented a study about VEGF and NGF in mares and foals from a physiologic delivery. It is carefully designed and well presented. Although I have some suggestions/corrections before it can proceed to publication. 

Abstract

Please include the general functions of VEGF and NGF that justify the performed study.

Introduction

Lines 68-69: Are the authors absolutely sure that it has never been reported?

Lines 103-104: Please include the full name of receptors

M&Ms

Please include the reference of the reagents/products used when it is applicable (ELISA Kits, PCR mix etc).

Lines 213-218: This is not a description of the table 1, but the description of PCR procedure. Please move this to the text and properly described what is shown in the table. Moreover, a gene table must have the full name of genes in italic bellow.

Results

Lines 247-248: It would be enriching to add illustrative images of the placental portions in supplementary file.

Discussion

Line 234: change decrease to decreased

Author Response

Responses to reviewers

Review 2

The authors presented a study about VEGF and NGF in mares and foals from a physiologic delivery. It is carefully designed and well presented. Although I have some suggestions/corrections before it can proceed to publication.

The authors thank the reviewer for his careful work and suggestions.

Abstract

Please include the general functions of VEGF and NGF that justify the performed study.

Done, thanks for the suggestion. A sentence was added to the beginning of the abstract, highlighted in the text.

Introduction

Lines 68-69: Are the authors absolutely sure that it has never been reported?

The authors are absolutely sure, thank you.

Lines 103-104: Please include the full name of receptors.

Done, thanks for the suggestion.

M&Ms

Please include the reference of the reagents/products used when it is applicable (ELISA Kits, PCR mix etc).

Reagent/product references used in ELISA and PCR assays were added in M&M section and highlighted in the text. Thanks for the suggestion.

Lines 213-218: This is not a description of the table 1, but the description of PCR procedure. Please move this to the text and properly described what is shown in the table. Moreover, a gene table must have the full name of genes in italic bellow.

All requested changes have been made and highlighted in the text and in Table 1, thanks for the suggestions.

Results

Lines 247-248: It would be enriching to add illustrative images of the placental portions in supplementary file.

The authors agree and thank for the suggestion, but the team of pathologists who conducted the histopathological examination of the placenta is not included among the authors and therefore we are not authorized to disclose the images of the fetal membranes.

Discussion

Line 324: change decrease to decreased

Done, thanks for the suggestion.